# Influences of Carbonated Recycled Concrete Fines on Cement Hydration

**Jiake Zhang** , **Liupeng Zhang** , **Boyang Xu** * and **Jie Yuan** *

The Key Laboratory of Road and Traffic Engineering, Tongji University, Ministry of Education,
Shanghai 201804, China; zhjiake@tongji.edu.cn (J.Z.); 2233437@tongji.edu.cn (L.Z.)
* Correspondence: 2133435@tongji.edu.cn (B.X.); yuanjie@tongji.edu.cn (J.Y.)

**Abstract:** The preparation of recycled concrete aggregate generates a lot of fines, which are obstacles for implementing the recycled concrete aggregate. In this work, carbonation treatment is applied to improve the properties of recycled concrete fine, and the influences of carbonated recycled concrete fine (CRCF) on cement hydration process are evaluated. Both fresh and hardened properties of the cement paste samples replacing 0 to 30% of the CRCF are measured. The results reveal that the addition of CRCF obviously accelerates the hydration process of cement, especially during the early stage, and the initial and final setting times of the cement paste containing 30% CRCF are both reduced by approximately 25% compared to the control. The CRCF improves the strength gain of cement, and that influence becomes obvious with longer curing; the relative compressive strength of cement paste containing 30% CRCF is increased by 18% relative to the control after being cured for 28 days. At the same time, the early hydration of cement paste is accelerated with the addition of CRCF and the total hydration heat after 48 h of cement paste is significantly decreased with the addition of CRCF. Specifically, the total hydration heat after 48 h of cement paste with 30% CRCF is less than 50% of that with 0% CRCF. Besides that, CRCF consumes CH in cement paste and improves the pore structure of hardened cement paste. The morphology of hydrated samples shows that the shape of ettringite formed within the control sample with 0% CRCF is longer than those of the other ones formed in cement paste with CRCF, and the length decreases as the CRCF contents increase. In addition, the sample containing 30% CRCF does not show the particles, which means that CRCF reduces the ettringite forming in hardened paste samples. Thus, the findings from this work provide a better understanding of the field of waste concrete reuse.

**Keywords:** carbonated recycled concrete fines; cement hydration; solid-phase composition; microstructure





## 1. Introduction

The use of recycling materials in concrete has gained attention in recent years due to its potential to reduce environmental impacts while enhancing the mechanical properties of concrete. The incorporation of waste materials, such as waste glass, coal bottom ash, and marble powder, as a replacement for traditional aggregates or cement can contribute to the sustainable development of the construction industry [1–6]. In addition, several researchers have investigated the effects of incorporating waste materials such as waste lathe scraps, recycled steel wires from waste tires, rubber tree seed shells, dispersed coconut fibers, and recycled PET into concrete on its properties [7–13]. The results have demonstrated that the incorporation of these recycled materials can not only improve the mechanical properties of concrete but also enable the recycling of these waste materials, thereby reducing their negative impact on the environment [14,15]. Therefore, it is important to explore and promote the use of recycling materials in concrete production for the development of sustainable and eco-friendly construction practices.

The demolition and upgrading of infrastructure will generate a lot of construction waste, it is reported that waste concrete makes up about 32% to 75% of the construction

waste [16]. Thus, the reutilization of waste concrete not only provides resources for construction materials, but also solves the problems of environment pollution [17]. Generally, it is considered that the application of construction waste as recycled aggregate is a potential way for the reutilization of waste concrete. However, high amounts of fines were produced during the process of recycled concrete aggregate production, such as crushing, grinding, and sieving. The produced fines generally constitute about 15% of the crushed concrete [18,19]. The reutilization of these recycled fines is an obstacle of reusing the waste concrete. It has been investigated that the application of the recycled concrete fine (RCF) as fine aggregate or filler in concrete may either reduce the mechanical properties of recycled concrete or introduce air pollution [20,21].

Generally, recycled concrete can be divided into three types according to particle size: Recycled Concrete Fine (RCF), Recycled Fine Aggregate (RFA), and Recycled Coarse Aggregate (RCA). The particle size of RCF is less than 0.15 mm and contains a large amount of hardened cement and unhydrated cement, which has a certain hydration reactivity [22–26]. Due to the small particle size and high calcium content in RCF, the application of RCF as supplementary cementitious materials has been widely considered. However, the hydration reactivity of RCF is much lower than those of Portland cement and other supplementary cementitious materials. Moreover, the porous structure of RCF increases the water adsorption of the RCF-added system, which results in a high-water requirement for recycled concrete with RCF to achieve sufficient workability [27,28].

In order to decrease the effects of recycled concrete aggregate and RCF on the properties of recycle aggregate concrete or concrete prepared with RCF and expand the scope of application, different approaches have been proposed for the treatment of recycled aggregate and RCF. Currently, researchers using carbon dioxide to treat recycled concrete found that the properties of carbonated recycled concrete were enhanced compared with the untreated one and this approach has a $CO_2$ capture potential [29–31]. Some researchers found that the modified carbonation process enhanced the micro-structure and surface texture of the RCA, resulting in improved properties, such as higher compressive strength, lower water absorption, and better resistance to freezing and thawing cycles [32,33]. The main component of recycled concrete that can be carbonated is the adhesive paste, which consists of calcium silica hydrate (C-S-H), calcium hydroxide (CH), calcium sulfoaluminate hydrates, and unhydrated clinker grains [34]. The reactions between $CO_2$ and CH and C-S-H in hardened paste are the primary carbonation reactions, and the reaction products mainly include $CaCO_3$ and silica gel [35–37]. Nevertheless, the primary components of RCF consist of crushed aggregate particles and partially hydrated cement paste, which include both unhydrated cement grains and hydrated products of cement [21]. Therefore, RCF can be carbonated as well.

Thus, it is necessary to investigate the influences of carbonated recycled concrete fine (CRCF) on cement hydration, the development of pore structure, and the mechanical properties of cement-based materials, which gives the feasibility of using the CRCF as an additive in recycled concrete. The aim of this work is to investigate the influences of different CRCF contents on both the fresh property and hardened property of cement paste, which indicates the interactions between CRCF and cement particles.

## 2. Raw Materials and Testing Methods

### 2.1. Raw Materials

In this study, recycled concrete fine (RCF) was obtained by grinding a hardened cement paste that was cured in 80 °C hot water for 30 days. The hardened cement paste had a water-to-binder (W/B) ratio of 0.35 and it was placed in an oven at 100 °C for two days before grinding. The 28-day compressive strength of the hardened paste sample was 53.6 MPa. The oven-dried hardened cement paste was placed in a ball grinder machine for about 30 min for grinding. The carbonated recycled concrete fines are shown in Figure 1.

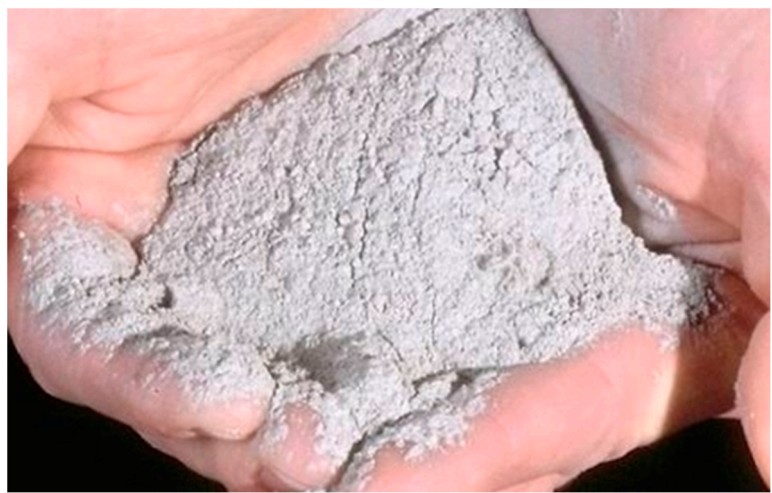

**Figure 1.** Carbonated recycled concrete fine.

Ordinary Portland cement (P·I 42.5) was applied in this study and the chemical composition of the cement is summarized in Table 1.

**Table 1.** Chemical composition of cement.

| Composition | CaO | $Al_2O_3$ | MgO | $Fe_2O_3$ | $SiO_2$ | $SO_3$ |
|---|---|---|---|---|---|---|
| Content (%) | 65.4 | 5.4 | 3.4 | 2.8 | 21.0 | 2.0 |

The particle size of the carbonated/uncarbonated RCF and ordinary Portland cement was measured using laser particle size analysis, as shown in Figure 2, and the particle size distributions of the raw materials are shown in Figure 3. Thermogravimetric analyses were conducted on both carbonated/uncarbonated RCF to identify the composition change during carbonation, as shown in Figure 4. The characteristic peak corresponding to $Ca(OH)_2$ disappeared for the carbonated RCF, which was associated with the weight loss from 410 to 430 °C on the TG curve. Based on the weight loss calculation, the $Ca(OH)_2$ content in the uncarbonated RCF was equal to 15.7%.

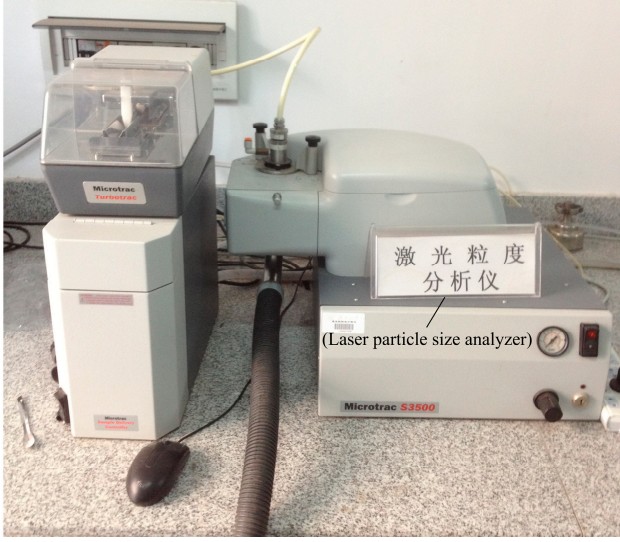

**Figure 2.** Laser particle size analysis.

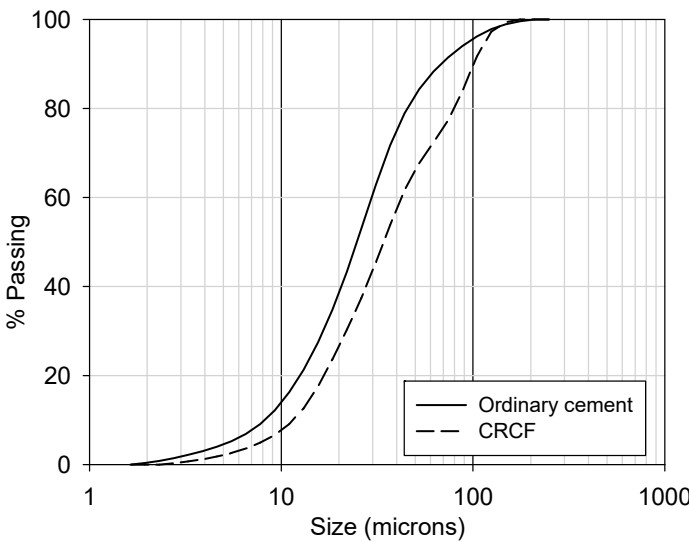

**Figure 3.** Particle size distributions of the original materials.

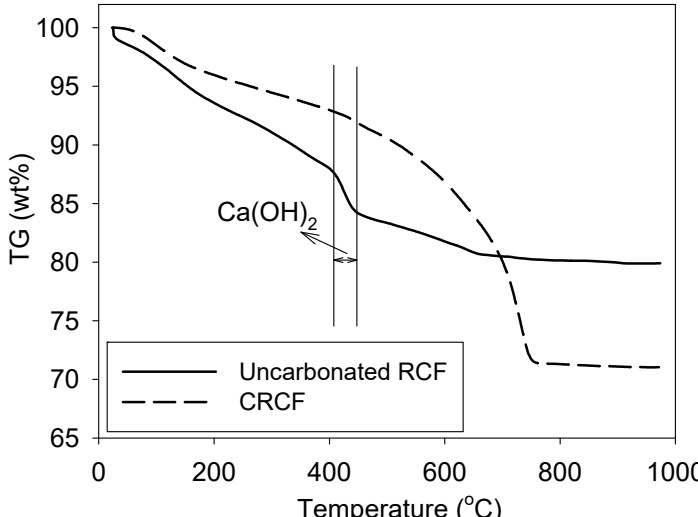

**Figure 4.** TG curves of the carbonated/uncarbonated recycled fine.

### 2.2. Testing Methods

#### 2.2.1. Carbonation

Recycled concrete fine (RCF) was placed in a carbonation chamber to carbonate for 7 days. The accelerated carbonation test was conducted in accordance with the Chinese standard GB/T 11974-1997 with the carbonation condition of T = 20 ± 2 °C, RH = 60 ± 5%, and $CO_2$ concentration = 20 ± 2%.

#### 2.2.2. Water Requirement and Setting Time

The water requirement of the paste samples containing 0% to 30% carbonated RCF with a 5% interval was determined as the paste samples reached the standard consistency. Tests were conducted in accordance with the Chinese standard GB/T 1346-2011. The standard consistency is obtained when the 10 mm needle of the Vicat apparatus penetrates 34 ± 1 mm into the paste. The water requirement (P) of the cement mixture equals the mass of mixing water divided by the mass of cement.

The setting time indicates the cement paste solidification in the early stage [34]. Paste samples were placed in a control curing room with RH > 90% and T = 20 after the water requirement test. A 1.13 ± 0.05 mm diameter needle attached to a 300 ± 1 g rod was

allowed to penetrate into the pastes every 15 min starting 30 min after molding. The initial setting time referred to the time of the penetration depth of the needle reaching $36 \pm 1$ mm. The time from the initial contact of cement and water until the penetration of the needle did not leave any indent on the cement paste surface, which was reported as the final setting time.

### 2.2.3. Flowability

The flowability of fresh paste samples was tested in accordance with the Chinese standard GB/T 8077-2000. A mini cone, which had a base diameter of 60 mm, a top diameter of 36 mm, and a height of 60 mm, was utilized in this work to characterize the flowability of the paste samples.

### 2.2.4. Compression Test

Fresh mortar mixtures, which were prepared with a W/B ratio of 0.35, were cast into $40 \times 40 \times 40$ mm cubic molds [38]. Samples were demolded after 24 h, and they were then placed in a curing room with T = $20 \pm 2$ °C and RH = 100% for 3 d, 7 d, and 28 d of curing since the mixing time for the compressive strength test. The test process is shown in Figure 5.

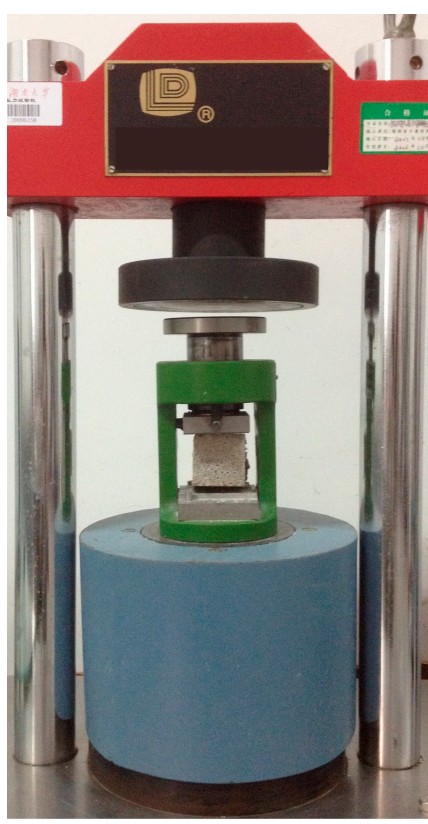

**Figure 5.** Compressive strength test.

### 2.2.5. Calorimetry

An isothermal calorimeter was utilized in this work to investigate the influences of CRCF on the hydration heat release during cement hydration. Samples containing 0% to 30% with a 5% interval of the CRCF were prepared, and the W/B ratio of 0.35 was selected in this work. The first 48 h measurements of each sample were analyzed.

### 2.2.6. Thermogravimetric Analysis (TGA)

About 10 mg of hardened paste samples were obtained from the center part of compression test specimens and grinded into powders with a diameter less than 0.08 mm. The

grinded powders were placed into a 100 °C oven until constant mass before conducting the thermogravimetric analysis. The temperature change was set at a rate of 10 °C/min from room temperature to 1000 °C. Both thermal gravimetric (TG) and differential scanning calorimetry (DSC) versus temperature curves were obtained for each sample. The thermogravimetric analysis process is shown in Figure 6.

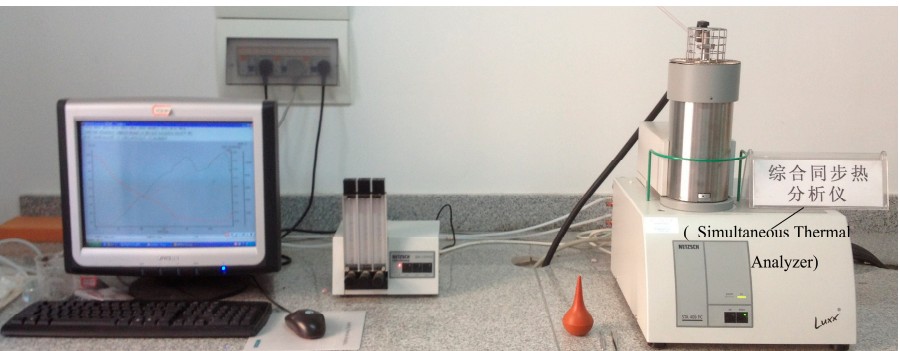

**Figure 6.** Thermogravimetric analysis process.

2.2.7. Porosity and Pore Size Distribution of Hardened Pastes

The porosity and pore size distribution of the paste samples were characterized using a mercury intrusion porosimeter (MIP) with a maximum mercury intrusion pressure of 210 MPa. The paste particles with a diameter around 0.5 cm were obtained from the cubic specimens for the compression test. The particle samples were vacuum-dried at 60 °C for 24 h before the MIP test. The pore diameter d at an applied pressure P was calculated using $d = -4\gamma \cos \theta/P$, where $\gamma = 0.483 \, \text{Mm}^{-1}$ is the surface tension of mercury; $\theta = 140°$ is the contact angle between mercury and the pore wall [39].

2.2.8. Microstructure Characterization

The paste particles with 1 cm$^3$ size were obtained from the middle part of the compressive-tested specimen and applied for the environmental scanning electron microscope (ESEM) measurement. The size and morphology and other hydration products can be determined at a given stage.

**3. Results and Discussion**

The effects of CRCF contents on the hydration heat release, mechanical property, solid-phase composition, and microstructure of the cement paste are discussed below.

*3.1. Water Requirement and Setting Time*

The water requirement of the cement paste with different replacement levels of CRCF to cement is shown in Figure 7. It can be seen from the figure that the water requirement of cement paste generally increases with the increase in replacement level of CRCF between 0% and 15%, and no significant differences are noticed with a CRCF content above 15%. The addition of CRCF generally changes the particle packing mode of the cement particles, which may change the water requirement in the pastes [40]. Moreover, the porous structure of recycled fines generally enhances the water adsorption of CRCF and results in the increased the value of the standard water requirement. However, both initial and final setting time decrease as the CRCF content increases (Figure 8). The addition of CRCF accelerates the hydration of cement, which consumes water and shortens the setting time. Moreover, the increased CRCF content reduces the cement content in the paste, which hydrates the cement grains more efficiently and also contributes to the shortened setting time of the paste samples. The results agree with those of similar studies [41,42].

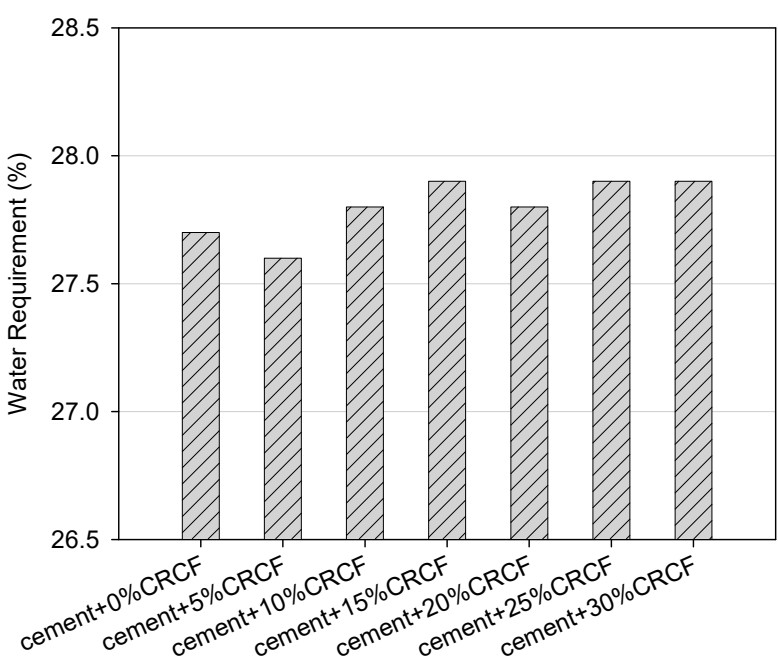

**Figure 7.** Water requirement of cement pastes with different CRCF contents.

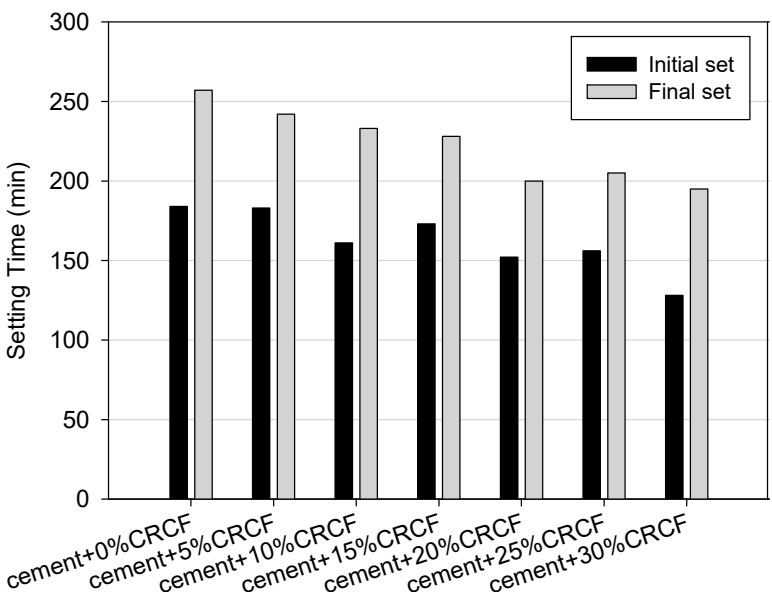

**Figure 8.** Initial and final setting time of cement samples with different CRCF contents.

### 3.2. Flowability

The influences of CRCF on the flowability of fresh cement paste were measured immediately after mixing. It can be seen that the flowability of fresh cement paste slightly decreases as the CRCF increases (Figure 9). It has been reported that the flowability of cement-based materials is affected by the W/B ratio, water evaporation rates, and hydration rates [43]. As the W/B ratio and water evaporation rates are the same in this work, it can be concluded that the decreased flowability is caused by the accelerated hydration rate. However, this effect is not significant when the CRCF dosage is lower than 15%.

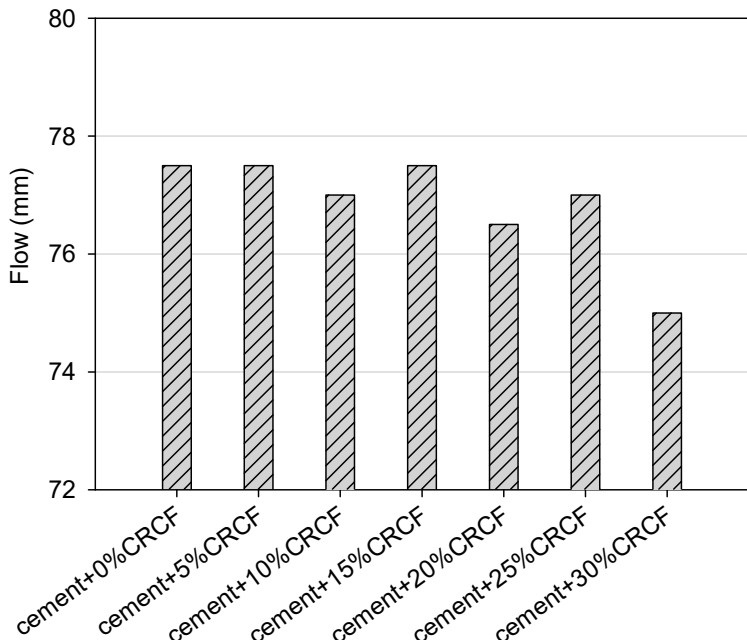

**Figure 9.** Effect of CRCF on the flow of fresh mixtures.

### 3.3. Hydration Heat

The heat evolution of cement paste specimens with different CRCF contents are shown in Figure 10, and the first heat flow peaks are observed about ten minutes after mixing with water. The first heat flow peak increases as the CRCF dosage increases. The carbonation product $CaCO_3$ in CRCF can rapidly react with $C_3A$ and $C_3S$ upon mixing and form monocarbonate hydrate ($Ca_4Al_2O_6 \bullet CO_3 \bullet 11H_2O$) and carbosilicate hydrate ($Ca_7(Si_6O_{18})(CO_3) \bullet 2H_2O$), respectively [44]. Moreover, the calcium carbonate particles can also be nucleation sites and accelerate the hydration of cement, especially for the hydration of $C_3A$ and $C_3S$, which can be confirmed by the larger heat release peak around 10 min after the mixing. The second heat flow peaks appear about 12 h after mixing.

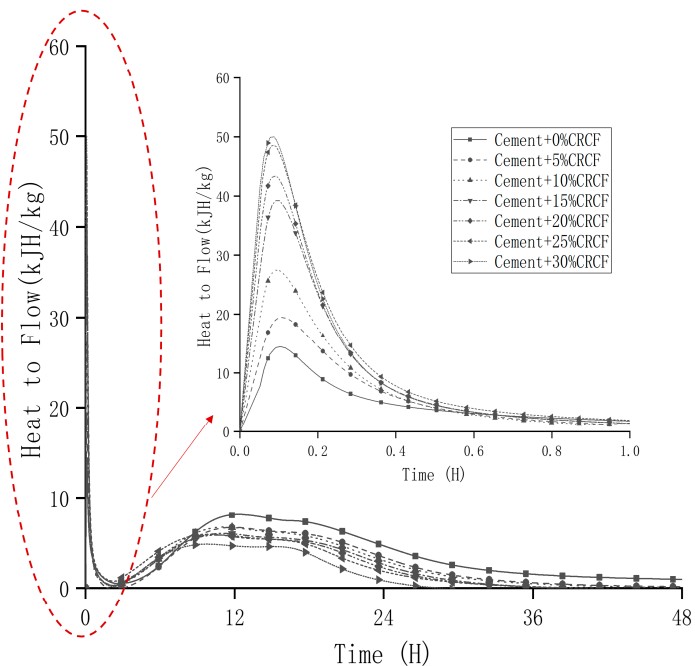

**Figure 10.** Heat evolution of cement with different CRCF contents.

The cumulated hydration heat curves of cement paste specimens with different replacement levels of CRCF are shown in Figure 11. It can also be clearly seen from the figure that the early hydration of cement paste is accelerated with the addition of CRCF. However, due to the reduced cement content, the total hydration heat after 48 h of cement paste is significantly decreased with the addition of CRCF. The results obtained in this study suggest that CRCF can effectively accelerate the early hydration rate of cement-based materials while the total hydration reaction and the content of hydration products are reduced. The decreased total hydration product can bring negative effects to the pore structure and mechanical properties of the cement paste with CRCF, which is discussed in the following sections [45–47].

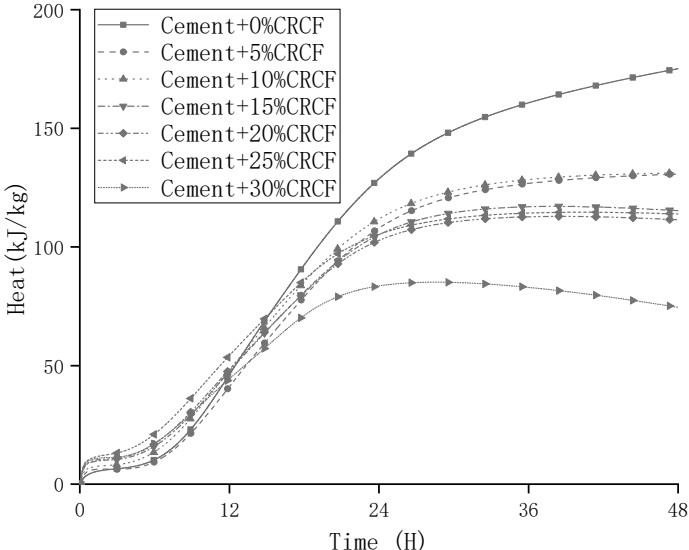

**Figure 11.** The cumulated heat of hydration of paste samples with different CRCF contents.

To further explore the influences of CRCF on cement hydration, the Krstulović–Dabić kinetics model was utilized in this research, which is the widely used hydration kinetic model [48,49]. The model divides the hydration process into three stages according to the hypothesis of nucleation and growth of hydration products and cement particle dissolution [50]. The total heat release of cement can be calculated based on the exothermic condition measured during the cement hydration process, and the cement hydration degree $\alpha$ and cement hydration rate $d\alpha/dt$ in different hydration periods can also be obtained [51]. Thus, the dynamic parameters can be obtained by substituting $\alpha$ and $d\alpha/dt$ into the Krstulović–Dabić model. After that, the relationship between the hydration rate and hydration degree at each stage can be obtained by substituting the dynamic parameters into the differential equations. Figure 12 shows the relationship between the hydration rate and hydration degree of the cement paste with different CRCF contents in each stage. The intersection of the simulated curves of $F_1(\alpha)$ and $F_3(\alpha)$ with $F_2(\alpha)$ ($\alpha_1$ and $\alpha_2$) represents the turning point of the dominant factor from nucleation and crystal growth (NG) to interactions at phase boundaries (I) and from I to diffusion (D), respectively. The value of $\alpha_1$ in the figure reflects the quality of nucleation and growth of hydration products. It can be seen from the figures that the value of $\alpha_1$ gradually increases with CRCF contents, which shows that the addition of CRCF can improve the hydration degree between the period of NG and I. However, such an influence is not that obvious compared with the influence of CRCF on $\alpha_2$, which indicates that CRCF extends the NG and I procedure and postpones the D procedure during cement hydration. In that case, CRCF improves the cement samples to form a homogeneous microstructure during hydration.

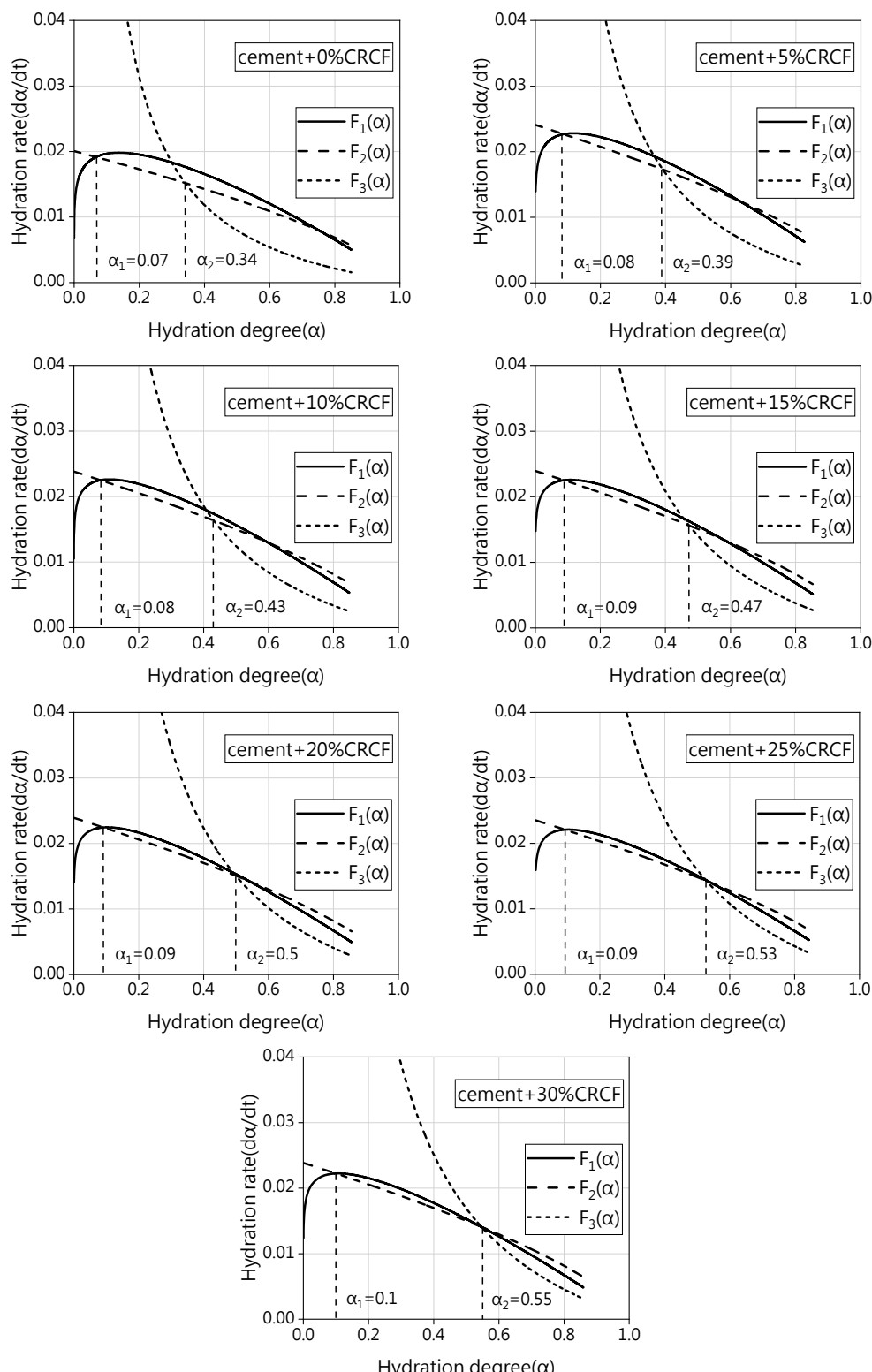

**Figure 12.** Relationships between hydration degree and hydration rate for paste samples with various CRCF contents.

The difference ($\Delta\alpha$) between $\alpha_1$ and $\alpha_2$ represents the degree of phase boundary reaction, and a greater $\Delta\alpha$ indicates a better phase boundary reaction. The relationship between $\Delta\alpha$ and the CRCF content is plotted in Figure 13. It can be noted from the figure that $\Delta\alpha$ gradually increases with the CRCF content, which suggests that CRCF improves

the cement phase boundary. Nevertheless, such an influence is reduced as the CRCF dosage increases. In the three stages of the hydration, both NG and I consist of fast hydration rates, which are significantly faster than the D process. Due to the influences of CRCF, both NG and I processes are accelerated, which also reveals that the addition of CRCF accelerates cement hydration and increases the heat release during the early stage.

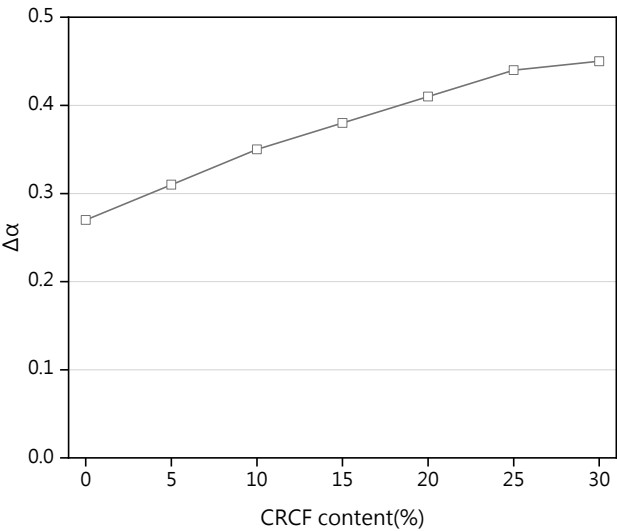

**Figure 13.** Relationship between $\Delta\alpha$ and CRCF content.

*3.4. Compressive Strength*

The compressive strength of cement pastes after 3, 7, and 28 days of curing are shown in Figure 14, and three samples were tested for each mix. The compressive strength of the hardened cement paste gradually decreases as the CRCF content increases for all three curing period samples, except for samples containing 20% CRCF. However, the decrease in compressive strength with the addition of CRCF is not that obvious. For cement paste with 30% CRCF, the 28-day compressive strength is just 18.1% lower than that of cement paste with 0% CRCF. With the increase in CRCF content, the decrease in cement content results in the decrease in hydration products and affects the pore structure of cement paste. By dividing the compressive strength by the cement content, we can obtain the relative compressive strength. The relative compression strength per cement is shown in Figure 15, and it can be noted that the relative compressive strength increases as the CRCF content increases, which can be attributed to the accelerated cement hydration due to the dilution effects from the replacement of CRCF. The increase in relative compressive strength with CRCF content is more obvious for samples with longer curing age. This indicates that the CRCF contributes little to the compressive strength at the early curing period, and the influence of CRCF in strength gain of the paste becomes stronger with longer curing time. The accelerated cement hydration may come from the reaction between the carbonation products of RCF and the cement hydration products, which results in the formation of the secondary hydration products. The main components of CRCF are calcite and aragonite [52], and these components can either promote the cement hydration and the growth of hydration products by the nucleation effects or act as a filler in the paste samples [53–55]. The influence of adding CRCF on the compressive strength of paste is similar to adding fly ash in the paste system [56,57].

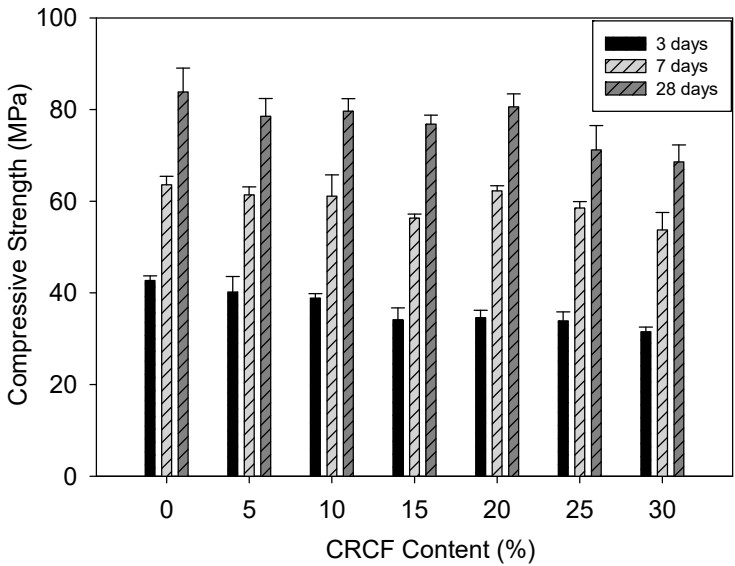

**Figure 14.** Compressive strength of cement samples with different CRCF contents.

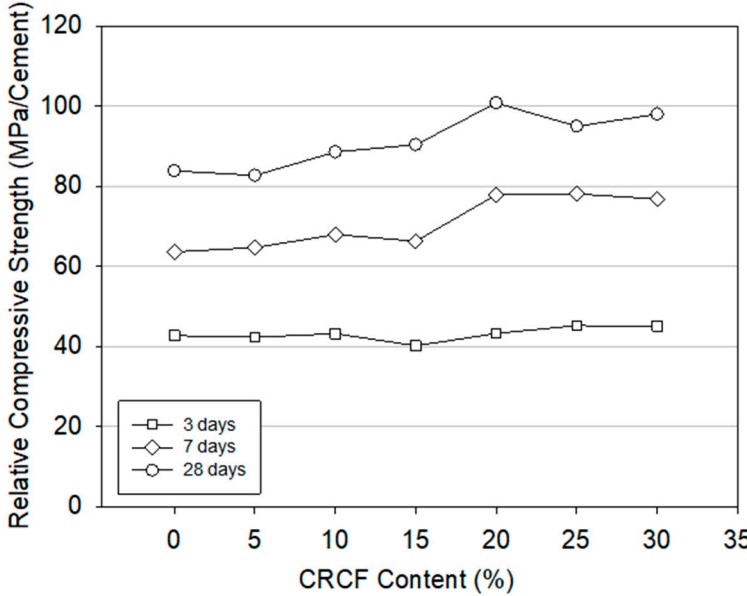

**Figure 15.** Compressive strength performance of CRCF.

### 3.5. Thermogravimetric Analysis (TGA)

The weight loss versus temperature relationships for paste samples that were cured for 3, 7, and 28 days during the TGA test are shown in Figure 16. The Ca(OH)$_2$ (CH) in the paste sample can be roughly quantified on the basis of the mass loss between 450 and 550 °C [54]. The CH content can be calculated based on the following equation [35]:

$$Ca(OH)_2 \rightarrow CaO + H_2O, \text{ and } CH\% = 4.11\, d_h$$

where $d_h$ is the mass loss of CH that can be estimated based on the TG data. Figure 17 shows the CH content of cement paste with different CRCF contents, and it clearly indicates that the CH content (%$w_t$ per gram of cement) gradually decreases with the increase in CRCF content. Generally, the replacement of CRCF to cement can generally enhance the hydration of cement due to the dilution effect. However, it can be obtained from the experimental results that CRCF in cement paste can consume CH in the system during cement hydration. The unhydrated cement within CRCF can react with CH and produce extra C-S-H gel with

the pozzolanic reaction. With the extended curing age, the continuous cement hydration results in a gradual increase in CH content.

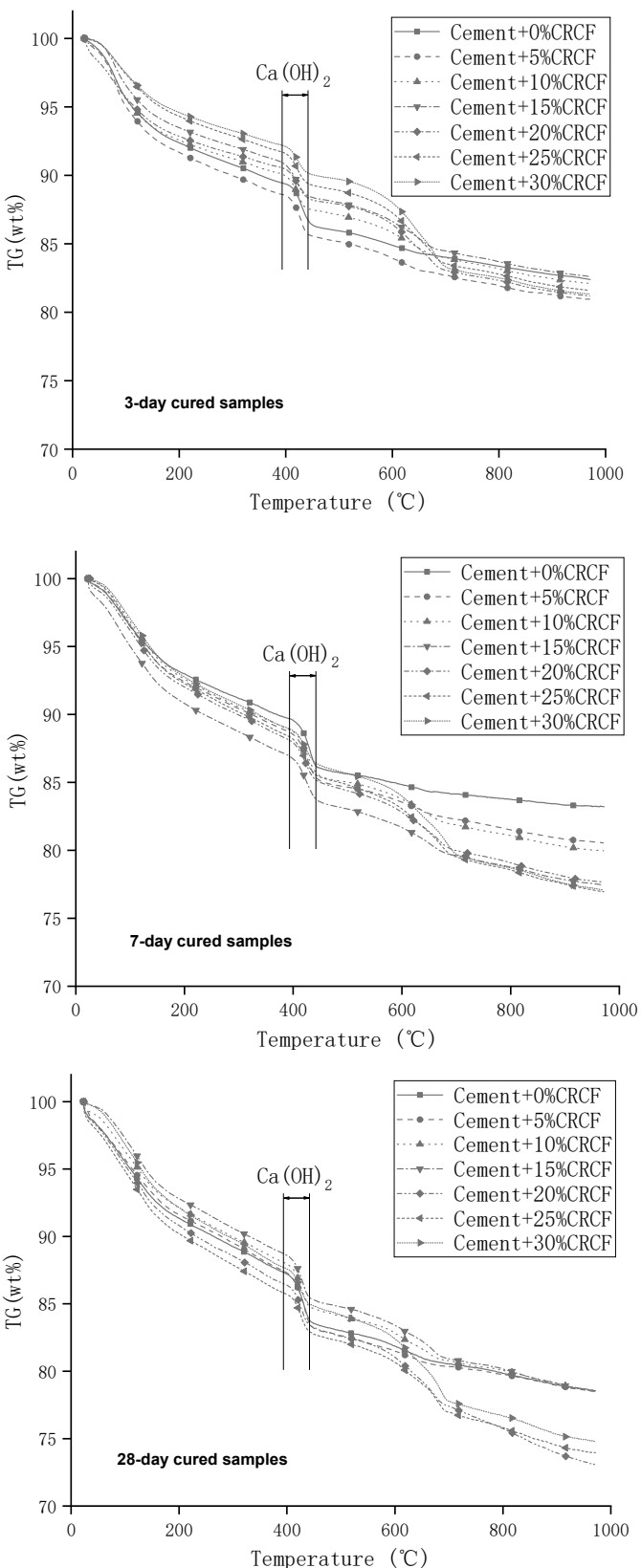

**Figure 16.** TG curves of the 3-day-, 7-day-, and 28-day-cured cement samples.

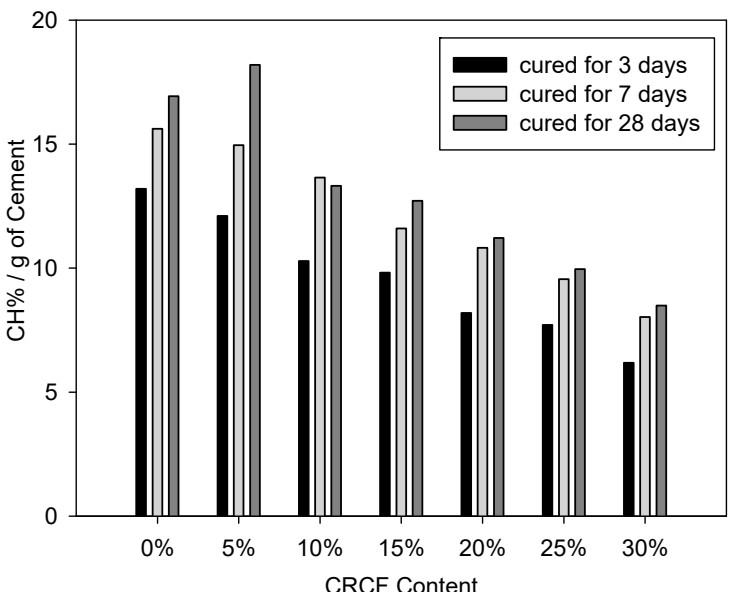

**Figure 17.** CH content in relation to CRCF contents.

### 3.6. Pore Size Distribution of Hardened Pastes

The hardened cement paste particle with a diameter around 0.5 cm was oven-dried under vacuum conditions before conducting the MIP test. As shown in Figure 18, the cumulative porosities of cement paste generally decrease as the CRCF content increases, except for the sample containing 10% CRCF. Cement pastes with 30% CRCF have the lowest porosity, which is about 15% lower than that of the cement paste without CRCF (Figure 19). The carbonation products of RCF are $CaCO_3$ and silica gel, which can either fill the voids of hardened cement paste or participate in cement hydration. The latter can be confirmed by the decreased CH content with the increase in CRCF content as shown by DTG analysis [58].

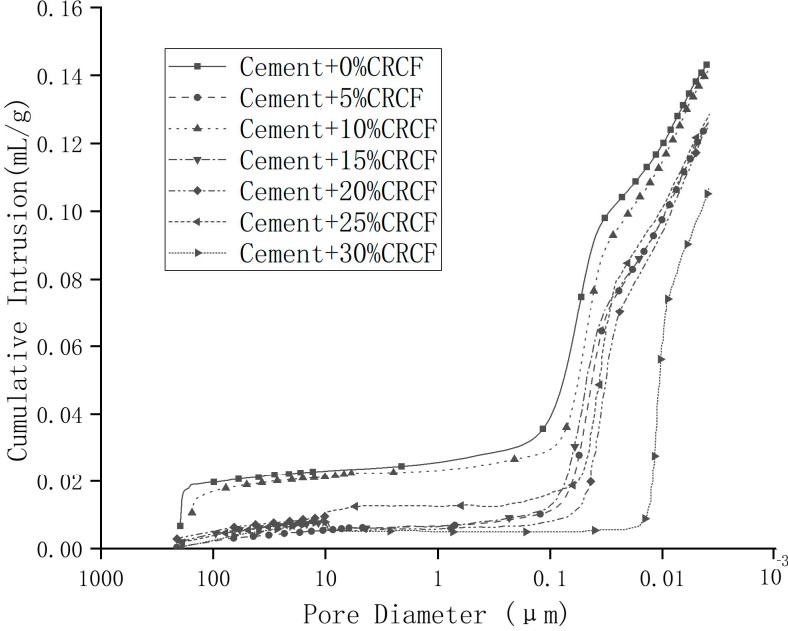

**Figure 18.** Cumulative mercury intrusion of the 7-day-cured hardened paste samples.

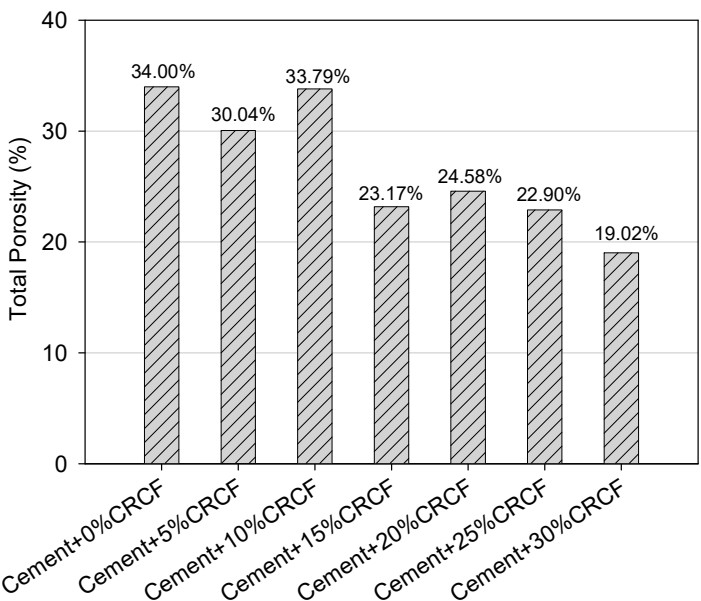

**Figure 19.** Total porosity of hardened cement pastes in relation to CRCF content.

However, it can also be seen from Figure 17 that the addition of CRCF increases the percentage of large pores within the cement paste. The increase in CRCF content decreases the total cement content in the system, which can lead to a decrease in total formed hydration products. The filler effects of CRCF effectively decreases the small pore volume, while the decreased hydration products result in the increase in the total volume of larger pores. The increased volume of large pores may be the reason for the decreased compressive strength of cement paste with the increase in CRCF content, as shown in Section 3.4 [59–61].

*3.7. Morphology*

Figures 20–23 show the morphology of the 7-day-cured cement paste with different CRCF contents at 2000× and 5000× magnitude. The shape of ettringite formed within the control sample with 0% CRCF is longer than those of the other ones formed in cement paste with CRCF, and the length decreases as the CRCF contents increase. The SEM images of the sample containing 30% CRCF do not show the "needle-like" particles, and such an observation indicates that CRCF can react with ettringite. It is noteworthy that the sample containing 30% CRCF shows a denser and less porous microstructure than the control sample, which is consistent with the result from MIP measurements [62,63].

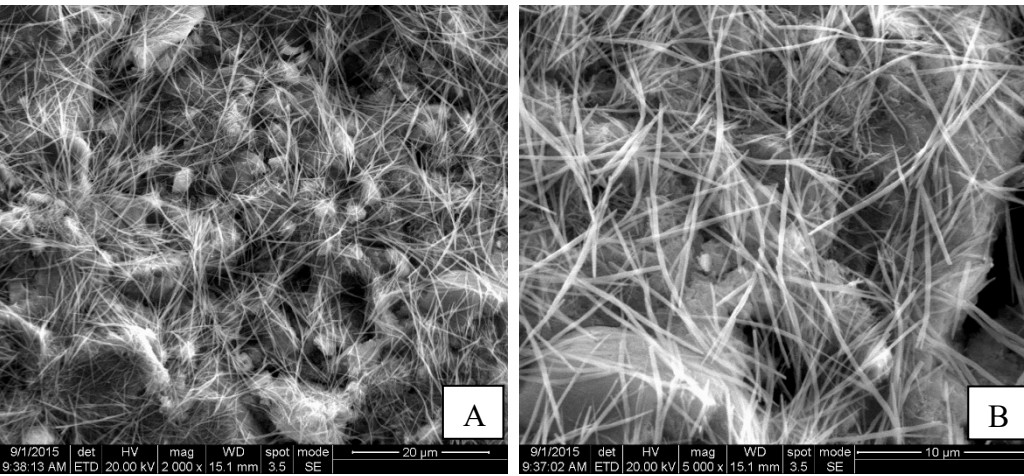

**Figure 20.** SEM images of 7-day-cured controlled sample: (**A**) 2000×; (**B**) 5000×.

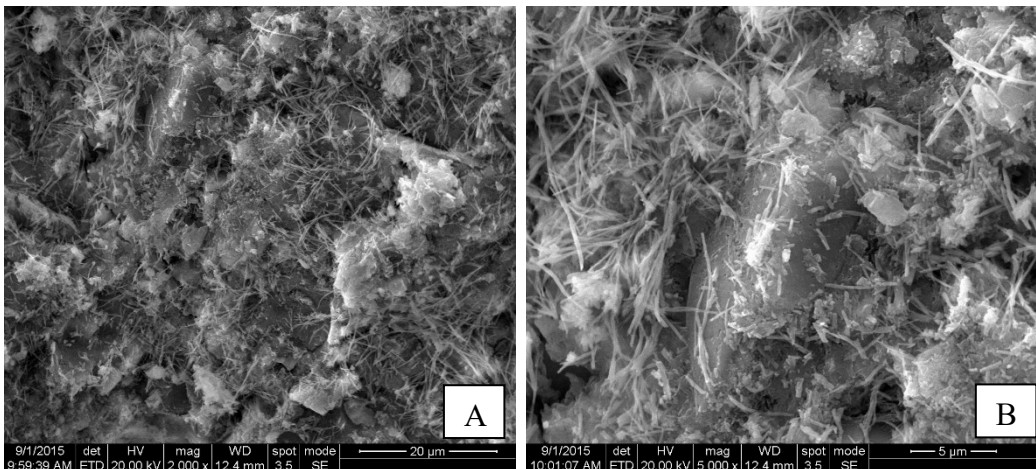

**Figure 21.** SEM images of 7-day-cured sample containing 10% CRCF: (**A**) 2000×; (**B**) 5000×.

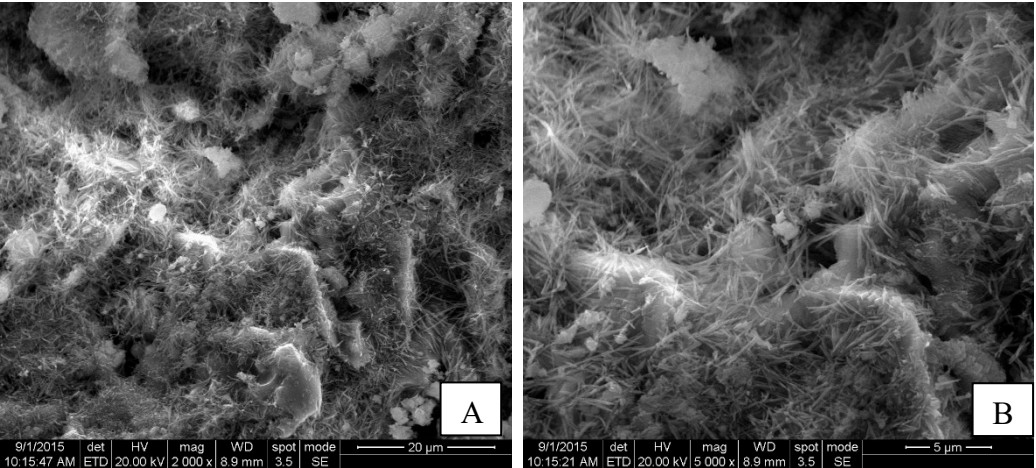

**Figure 22.** SEM images of 7-day-cured sample containing 20% CRCF: (**A**) 2000×; (**B**) 5000×.

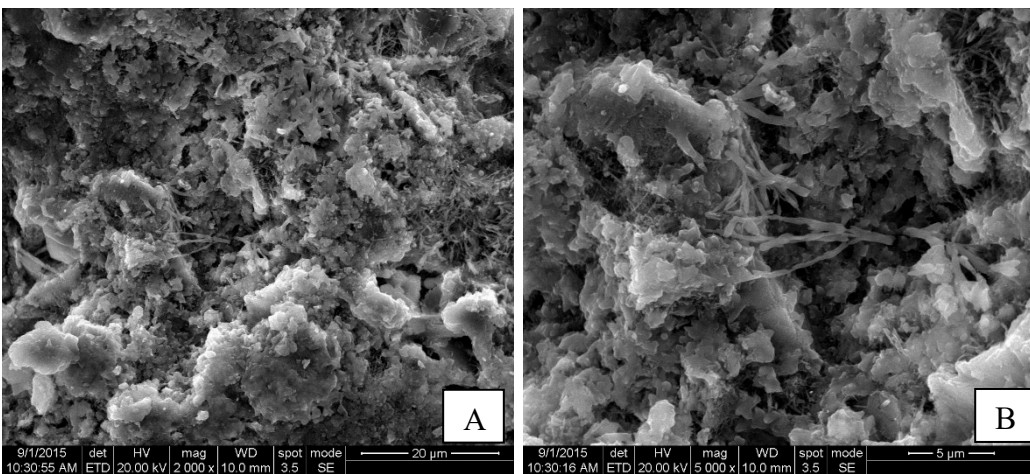

**Figure 23.** SEM images of 7-day-cured sample containing 30% CRCF: (**A**) 2000×; (**B**) 5000×.

## 4. Conclusions

This work investigated the influences of carbonated recycled concrete fine (CRCF) on cement hydration as the CRCF was added in cement paste. Both fresh properties and hardened properties of the cement paste with different CRCF contents were measured. Based on the experimental results, the following conclusions can be drawn:

(1) The CRCF reduces the setting time and flowability of fresh paste, and the initial and final setting times of the cement paste containing 30% CRCF both reduce by approximately 25% compared to the control. However, CRCF increases the water requirement in general. This indicates that the CRCF can accelerate cement hydration after mixing with fresh paste, which can be revealed from the flowability measurement as well.

(2) The compressive strength of hardened paste decreases as the CRCF content increases due to the reduced cement content in samples. However, the relative compressive strength per cement increases as the CRCF content increases, except for the 3-day-cured sample with 15% CRCF, and the relative compressive strength of cement slurry containing 30% CRCF increases by 18% relative to the control after being cured for 28 days. The influence of CRCF on the strength gain of cement is similar to fly ash.

(3) The hydration heat analysis of cement samples with different CRCF dosages indicate that CRCF can reduce the total cement hydration heat and improve NG and I processes at the same time, which accelerates the cement hydration rate and increases the early hydration heat release.

(4) CH in the hydrated paste samples decreases as CRCF increases, which means that CRCF consumes CH in the cement paste. This indicates that the carbonated RF may introduce a pozzolanic reaction when it is mixed with cement paste.

(5) The porosity of the hardened paste samples generally deceases as the CRCF content increases, which illustrates that adding CRCF improves the pore structure of cement paste.

(6) The SEM images show that the ettringite formed in the hydrated paste becomes shorter and smaller as the CRCF content increases. Thus, CRCF indeed changes the morphology of hydrated cement paste.

This research converts the powder waste generated during the crushing process of recycled aggregates into a valuable resource, which holds significant importance for enhancing the utilization efficiency of recycled aggregates and promoting sustainable development.

**Author Contributions:** Conceptualization, J.Y. and J.Z.; methodology, J.Z., B.X. and J.Y.; data curation, J.Z., B.X. and L.Z.; software, J.Z. and L.Z.; formal analysis, J.Z. and L.Z.; resources, B.X. and J.Y.; writing—original draft preparation, J.Z.; writing—review and editing, J.Z., B.X. and J.Y; funding acquisition, J.Y. All authors have read and agreed to the published version of the manuscript.

**Funding:** The National Science Foundation of China, grant number U1933116.

**Institutional Review Board Statement:** Not applicable.

**Informed Consent Statement:** Not applicable.

**Data Availability Statement:** Not applicable.

**Conflicts of Interest:** The authors declare no conflict of interest.

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
