# Peer review of "Influences of Carbonated Recycled Concrete Fines on Cement Hydration"

_buildings, doi:10.3390/buildings13040926_

Round 1

Reviewer 2 Report

The authors uinvestigated CRCF in cement. The idea is good but needs to be improved.

Add results in terms % in abstract

Add a paragraph into introduction about importance of the use of recycling materials into concrete. For this purposes following studies can be utilized: influence of replacing cement with waste glass on mechanical properties of concrete; use of recycled coal bottom ash in reinforced concrete beams as replacement for aggregate; concrete containing waste glass as an environmentally friendly aggregate: a review on fresh and mechanical characteristics; mechanical behavior of crushed waste glass as replacement of aggregates;flexural behavior of reinforced concrete beams using waste marble powder towards application of sustainable concrete

Why RCF is less than 0.16 mm? Where did you find this information?

There are also smiliar studies. What is your difference? Include novelty.

Check Error! Reference source not found.. 73..

40x40x40 mm is very very small samples. Which standard recommended this dimension?

How this recycled materials for this study is obtained?

Compressive strength seems to very high. What is the reason? Justify your results with literature

The reason for selecting design mixture should be added.

Compare your results with existing studies

Add photos for test setup?

Add photos for utilized materials. There is no photo related to which materials are utilized.

The importance of recycling materials to overcome enviermental prbolem should be added to introduction using:  improvement in bending performance of reinforced concrete beams produced with waste lathe scraps; performance assessment of fiber-reinforced concrete produced with waste lathe fibers; performance evaluation of fiber-reinforced concretes produced with steel fibers extracted from waste tire; investigation on improvement in shear performance of reinforced-concrete beams produced with recycled steel wires from waste tires; composition component influence on concrete properties with the additive of rubber tree seed shells; normal-weight concrete with improved stress–strain characteristics reinforced with dispersed coconut fibers; investigation of the physical-mechanical properties and durability of high-strength concrete with recycled pet as a partial replacement for fine aggregates; effects of waste powder, fine and coarse marble agregates on concrete compressive strength.

Please add damaged photos damaged photos of samples

Add recent studies on this subject to introduction. There are many studies on the introduction for this topic.

Conclusion should be improved. The recommendation consdiering all test should be given for engineers.

Reviewer 3 Report

1)      “In this work, carbonation treatment was applied 9 to improve the properties of recycled concrete fine and the influences of carbonated recycled concrete fines (CRCF) on cement hydration process were evaluated.” Review sentence.

2)      “cement paste samples which contain 0 to 30% of the CRCF were measured” Used as a replacement for cement? Make it clear in the abstract.

3)      “Currently, researchers using carbon dioxide to treat recycled concrete and found that the properties of carbonated recycled concrete were enhanced compared with the untreated one [7,8].” Also, this approach has a CO2 capture potential. Please see the reference: https://doi.org/10.1016/j.conbuildmat.2022.126357

4)      “Ordinary Portland cement (P·I 42.5) was applied in this study and the chemical com- position of the cement is summarized in Error! Reference source not found..” Check the source error.

5)      Enter the chemical composition of the CRCF

6)      Insert a table with the evaluated compositions in the materials and methods section. Was CRCF used as a replacement for cement?

7)      “The SEM images of the sample contained 30% CRCF does not show the “needle like” particles, and such observation indicates that CRCF can react with ettringite.” Insert reference to support this statement. Existing studies in the literature reported similar results? How does this reduction in ettringite crystals affect mechanical performance, for example?

8)      Include a final sentence in conclusion, highlighting the study's contribution and suggestions for further research in the area.

Round 2

Reviewer 1 Report

·       In the present paper, the authors have studied experimentally, The effect of fiber volume fraction on fiber distribution in steel fiber-reinforced self-compacting concrete

·       The outcome of this research is significant in providing useful information about the effects of fibers on self-compacted concrete.

The authors answered all the comments and suggestions were considered

Author Response

Thank you for your kind work.

Reviewer 2 Report

The authors improved the quality of paper. But some points did not answered well.

Point 1: The authors added only for compressive strength and setting time. What about the others?

Point 2 and 13. The authors ignored the suggested ones. Please revise this section including the suggested ones

Point 3: The answer did not included in paper.

Point 6: So you utilized damaged samples for compressive strength. Is this applicable?

Point 9: The authors just explained reason of CRCF. What about W/B?

Point 14: It should be good to add some damaged ones

Reviewer 3 Report

Accept

Author Response

Thank you for your kind work.

Round 3

Reviewer 2 Report

The paper can be accepted.